# Integrated silicon qubit platform with single-spin addressability, exchange control and single-shot singlet-triplet readout

M. A. Fogarty[1,6], K. W. Chan [1], B. Hensen[1], W. Huang[1], T. Tanttu[1], C. H. Yang[1], A. Laucht[1], M. Veldhorst[2], F. E. Hudson[1], K. M. Itoh[3], D. Culcer [4], T. D. Ladd[5], A. Morello [1] & A. S. Dzurak[1]

Silicon quantum dot spin qubits provide a promising platform for large-scale quantum computation because of their compatibility with conventional CMOS manufacturing and the long coherence times accessible using $^{28}$Si enriched material. A scalable error-corrected quantum processor, however, will require control of many qubits in parallel, while performing error detection across the constituent qubits. Spin resonance techniques are a convenient path to parallel two-axis control, while Pauli spin blockade can be used to realize local parity measurements for error detection. Despite this, silicon qubit implementations have so far focused on either single-spin resonance control, or control and measurement via voltage-pulse detuning in the two-spin singlet–triplet basis, but not both simultaneously. Here, we demonstrate an integrated device platform incorporating a silicon metal-oxide-semiconductor double quantum dot that is capable of single-spin addressing and control via electron spin resonance, combined with high-fidelity spin readout in the singlet-triplet basis.

[1] Centre for Quantum Computation and Communication Technology, School of Electrical Engineering and Telecommunications, The University of New South Wales, Sydney, NSW 2052, Australia. [2] QuTech and Kavli Institute of Nanoscience, TU Delft, Lorentzweg 1, 2628CJ Delft, The Netherlands. [3] School of Fundamental Science and Technology, Keio University, 3-14-1 Hiyoshi, Kohoku-ku, Yokohama 223-8522, Japan. [4] School of Physics, The University of New South Wales, Sydney 2052, Australia. [5] HRL Laboratories, LLC, 3011 Malibu Canyon Rd., Malibu, CA 90265, USA. [6] Present address: London Centre for Nanotechnology, UCL, 17-19 Gordon St, London WC1H 0AH, UK. Correspondence and requests for materials should be addressed to M.A.F. (email: m.fogarty@ucl.ac.uk) or to B.H. (email: b.hensen@unsw.edu.au) or to A.S.D. (email: a.dzurak@unsw.edu.au)

The manipulation of single-spin qubits in silicon, using either ac magnetic[1,2] or electric[3–6] fields at microwave frequencies, has been a powerful driver of progress in the field of solid state qubit development, in part due to the sophistication of microwave technology which allows convenient two-axis control of the qubit via simple phase adjustment, and the generation of complex pulse sequences for dynamical decoupling. This has resulted in high-fidelity single-qubit gates[2,4–7] and initial two-qubit gates now realised in a variety of structures[8–10]. To date, all demonstrations of single-shot readout in silicon systems employing spin resonance[1–4,6] have utilized single-spin selective tunnelling to a reservoir[11]. While convenient, this reservoir-based readout approach is not well suited to gate-based dispersive sensing[12], which has significant advantages in terms of minimizing electrode overheads for large-scale qubit architectures. In contrast, readout based on Pauli spin blockade[13] in the singlet–triplet basis of a double QD[14] is compatible with dispersive sensing and, when combined with an ancilla qubit, can be used for parity readout in quantum error detection and correction codes[15–17]. Moreover, because singlet–triplet readout can provide high-fidelity spin readout at much lower magnetic fields than single-spin reservoir-based readout[11], it allows spin-resonance control to be performed at lower microwave frequencies, which will benefit scalability.

Qubits based on singlet–triplet spin states were first demonstrated in GaAs heterostructures[14,18] and have now been operated in a variety of silicon-based structures[19–23]. High-fidelity single-shot singlet-triplet readout has also recently been demonstrated in various silicon systems[22,24,25].

Here, in order to combine the ability to address individual spin qubits using ESR with the voltage-pulse-based detuning control and high-fidelity readout of pairs of spins in the singlet-triplet basis, we employ a $^{28}$Si metal-oxide-semiconductor (SiMOS) double quantum dot device[26,27] (Fig. 1a, b) with a microwave transmission line that can be used to supply ESR pulses, similar to one previously used for demonstration of a two-qubit logic gate[8]. The device also includes an integrated single-electron-transistor (SET) sensor to achieve the single-charge sensitivity required for singlet–triplet readout. Electrons are populated into the two quantum dots (QD1 and QD2) with occupancy ($N_1$, $N_2$) using positive voltages on gates G1 and G2. An electron reservoir is induced beneath the Si–SiO$_2$ interface via a positive bias on gate ST, which also serves as the SET top gate. The reservoir is isolated from QD1 and QD2 by a barrier gate B (see Fig. 1a, b).

## Results

**Single-shot singlet–triplet readout.** Figure 1c shows the stability diagram of the double QD system in the charge regions ($N_1$, $N_2$) where we operate the device. When two electrons occupy a double quantum dot, the exchange interaction results in an energy splitting between the singlet ($S$) and triplet ($T_-$, $T_0$, $T_+$) spin states. The exchange interaction can be controlled by electrical pulsing on nearby gates, providing a means to initialize, control and read out the singlet and triplet states[14]. At the core of singlet–triplet spin readout is the observation of Pauli spin-blockade (PSB)[19,28–31]. When pulsing from the (1, 1)→ to (0, 2) charge configurations, the QD1 electron tunnels to QD2 only when the two spatially separated electrons were initially in the singlet spin configuration. The triplet states are blockaded from tunnelling due to the large exchange interaction in the (0, 2) charge configuration. The blockade is made observable on the stability diagram by applying a pulse sequence[19,28] to gates G1 and G2 as depicted in Fig. 1c. After first flushing the system of a QD1 electron to create the (0, 1) state at A, a (1, 1) state at B loads a randomly configured mixture of singlet and triplet states (solid

arrow in Fig. 1c). The current through the nearby single-electron-transistor (SET) is recorded at this position, tuned to be at the half-maximum point of a Coulomb peak. The system is then ramped to a variable measurement point (dashed arrows in Fig. 1c, d) where the SET current is measured again. A map of the comparison current $\Delta I_{SET}$ between these two points is created, where the derivative in sweep direction $d(\Delta I_{SET})/d(\Delta V_{G1})$ (Fig. 1c) decorrelates the capacitive coupling of the control gates to the SET island. A change in the charge configuration marks a shift in the SET current, clearly observed as bright/dark bands. The bright band in the centre of the (1,1)–(0,2) anti-crossing of Fig. 1c is consistent with PSB, where the blockade triangle is restricted to a narrow trapezoidal area, bounded by state co-tunnelling via the reservoir and the first available excited triplet state[19].

The charge sensor design used (Fig. 1a) is relatively insensitive to inter-dot charge transitions, due to the symmetry of the QD1 and QD2 locations with respect to the SET island[32]. In order to enhance the blockade signal for this layout, we employ state-latching using the nearby electron reservoir[33]. Recent studies of reservoir charge state latching[22,24] and intermediate excited states[34] in semiconductor quantum dot devices have led to methods to reduce readout error by almost an order of magnitude[22]. A variant of this state latching is observed and utilized here.

The latching is produced via asymmetric couplings of the two dots to the common electron reservoir[33], where a (1, 1)-(1, 2) dot-reservoir metastable charge state is produced via a combination of the low tunnel rate between QD2 and the reservoir (shown as $\Gamma_{Slow}$ in Fig. 1b) and co-tunnelling between QD1, QD2 and the reservoir ($\Gamma_{Fast}$ in Fig. 1b). The latching results in the prominent feature observed at the (1, 1)-(1, 2) transition in Fig. 1c. In contrast, when the system is initialized in the (0, 2) charge configuration, the singlet state is prepared robustly due to large energy splitting, and the resulting map in Fig. 1d has no latched PSB region, as expected. The energy splitting between the (0, 2) singlet ground state and first available triplet state is measured to be $(1.7 \pm 0.2)\%$ of the charging energy $E_C$ (see Supplementary Notes 1 and 2, Supplementary Figs. 1, 2). Typically for this device design $E_C \sim 10$–20 meV[35], indicating that this splitting exceeds electron thermal energies by two orders of magnitude. The first available triplet here is likely the first excited valley state[35–37]. To compare the visibility of the standard PSB and latched PSB, histograms of $\Delta I_{SET}$ are shown in Fig. 1f, g respectively. We find that state latching increases our measurement visibility from around 70 to 98%, reducing the misidentification error by more than 16-fold for this SiMOS device layout (see Supplementary Note 1 and Supplementary Fig. 3). We note that this measurement fidelity of $F_M = 99\%$ does not include errors that occur during the evolution from a separated (1, 1) charge state to the blockade region, which we discuss in more detail below.

**Singlet–triplet Hamiltonian for Silicon-MOS qubits.** The large valley splitting in SiMOS devices[8,35] allows us to restrict ourselves to the lowest valley state when considering spin dynamics near the (0, 2)–(1, 1) anti-crossing, which we now address. These dynamics are governed by a Hamiltonian in which single-spin distinguishability and exchange are in competition. Single-spin distinguishability arises from the varying Zeeman energy between each dot, interpreted as a site-specific effective $g$-factor and resulting in an energy difference $\delta E_Z = g_2 \mu_B B_2^z - g_1 \mu_B B_1^z$. For high in-plane magnetic field, the varying effective $g$-factors result from a combination of interface spin-orbit terms, which depend on local strain, electric fields, and the atomistic details of the oxide interface[35,38,39]. Further, recent studies have shown that this

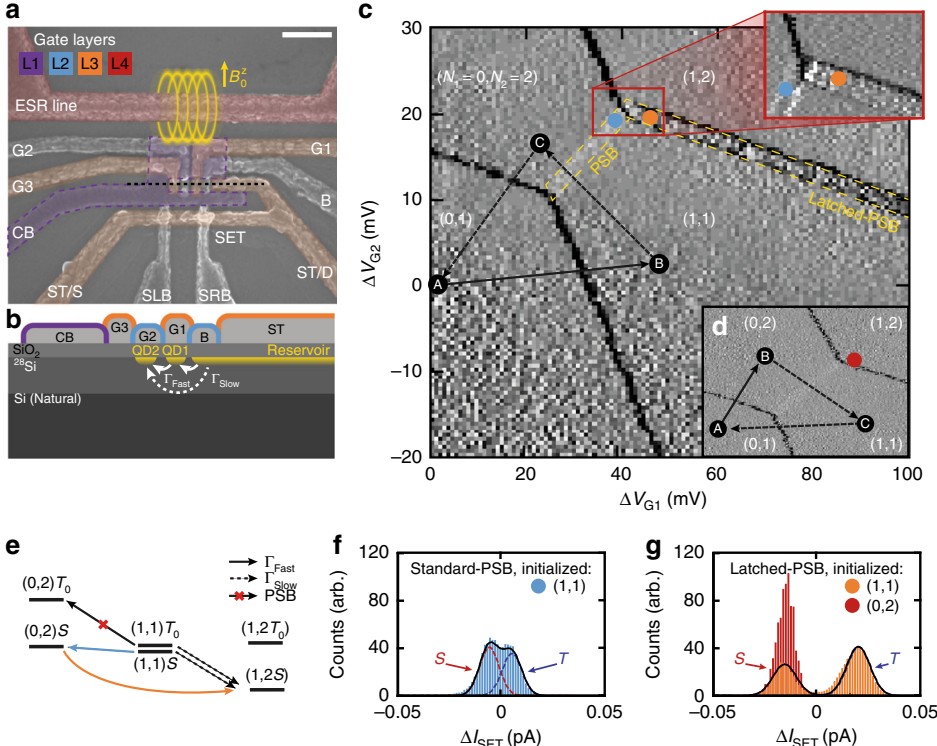

**Fig. 1** Silicon double quantum dot with latched Pauli spin blockade readout. **a** False-colored scanning electron micrograph of the device architecture. Dots are created under G1 and G2 electrodes and situated in the centre of the confinement gap. Scale bar is 200 nm. **b** Cross section illustrating dots under G1 and G2, which are tunnel-coupled with fast and slow tunnel rates $\Gamma_{Fast}$ and $\Gamma_{Slow}$ to an electron reservoir under gate ST. This reservoir is located on the drain (D) side of the sensor. **c** Cyclic pulsing[19,28] (arrows) through sequence A(0, 1)-B(1, 1)-C, where the location of point C is rastered to form the image, reveals latched spin blockade features (orange dot & top zoom-in). Shown is the differential transconductance d($\Delta I_{SET}$)/d($\Delta V_{G1}$), where $\Delta I_{SET}$ is the difference in SET current recorded at points B and C. **d** When point B lies in the (0, 2) charge region, no blockade is observed, as expected for an initial singlet state. **e** Observation of state-latching of the G2 dot is due to weak coupling to the reservoir. In order to populate the (1, 2) state, the existing (1, 1) state must co-tunnel via (0, 2) where PSB exists. If the state is not blocked (i.e. the $|S\rangle$ state) then an electron is free to tunnel from the reservoir to fill G1. Otherwise, the tunnelling from the reservoir is blocked, resulting in a spin-to-reservoir charge state conversion. **f** Histogram of $\Delta I_{SET}$ recorded at Standard-PSB readout location indicated by the blue marker on map **d**. **g** Histogram of $\Delta I_{SET}$ recorded at Latched-PSB readout location for B(1, 1) (orange) and B(0, 2) (red); there is a clear increase in sensitivity provided by the Latched-PSB readout

Hamiltonian parameter can be modulated by the direction of the applied field with respect to the crystallographic axis[23]. In previous devices we have observed g-factor differences between QDs as large as 0.5%[8]; at high-field, Overhauser contributions to $\delta E_Z$ are negligible in isotopically purified samples. At lower magnetic fields, magnetic screening from the superconducting aluminum gates may also contribute significantly to $\delta E_Z$[40]. For these experiments, the scale of $\delta E_Z$ is predominantly set by choice in the magnitude of the external magnetic field; however, inevitable microscopic dot-to-dot variation in this parameter for future qubit arrays would best be handled by calibration and refocusing methods[15].

The Hamiltonian term in competition with the Zeeman gradient is kinetic exchange, which lowers the energy of the spin singlet energy by an amount $J(\varepsilon)$ due to interdot tunnelling. In the standard Fermi-Hubbard model, $J(\varepsilon)$ is proportional to $2t_c^2(\varepsilon)/|\varepsilon|$ for large $\varepsilon$, where $t_c(\varepsilon)$ is the inter-dot tunnel coupling and $\varepsilon$ combines the on-site charging energy and electrochemical potential difference between the two dots[41] (Supplementary Note 3). In previous experiments[8] on a similar SiMOS two-qubit device the tunnel coupling at the anti-crossing was estimated as $900\sqrt{2}$ MHz. For both devices, $t_c$ at the anti-crossing is fixed, set by the device geometry. This parameter can be made tunable via the incorporation of exchange gates[42] into the SiMOS architecture. In this model, the ground-state singlet is hybridized between (0, 2) and (1, 1) charge states as $|S_H\rangle = \cos(\theta/2)|(1, 1)S\rangle +$

$\sin(\theta/2)|(0, 2)S\rangle$, where $\theta = -\tan^{-1}(2t_c/\varepsilon)$. Again neglecting higher energy valley or orbital states, the spin-triplet states $|T_m\rangle$ with two-spin angular momentum projection $m = 0, \pm 1$ are fully separated in the (1, 1) charge state. Besides being split from the $m = 0$ states by the mean Zeeman energy, $\bar{E}_Z$, the $|T_\pm\rangle$ states may couple to the hybridized singlet states by local magnetic fields which are orthogonal to the average applied field, as well as by spin orbit coupling. We summarize such terms by a spin-flipping term $\Delta(\theta)$.

Hence in the basis $\{|T_+\rangle, |T_0\rangle, |T_-\rangle, |S_H\rangle\}$, the approximate effective Hamiltonian is written (Supplementary Note 3)

$$H = \begin{pmatrix} \bar{E}_Z - \varepsilon/2 & 0 & 0 & \Delta(\theta) \\ 0 & -\varepsilon/2 & 0 & \delta E_Z \cos\theta \\ 0 & 0 & -\bar{E}_Z - \varepsilon/2 & -\Delta(\theta) \\ \Delta(\theta)^* & \delta E_Z \cos\theta & -\Delta(\theta)^* & \varepsilon/2 - J(\varepsilon) \end{pmatrix}. \quad (1)$$

A typical energy spectrum of this Hamiltonian as a function of detuning $\varepsilon$ is shown in Fig. 2a for small magnetic fields, $B^z \sim J(\varepsilon)/g\mu_B$.

**Characterizing the singlet-triplet Hamiltonian.** The anti-crossing between the $|S_H\rangle$ and $|T_\pm\rangle$ states due to $\Delta(\theta)$ can be used to map out the energy separation $\left|E_{S_H}(\varepsilon) - E_{T_-}(\varepsilon)\right|$ as a

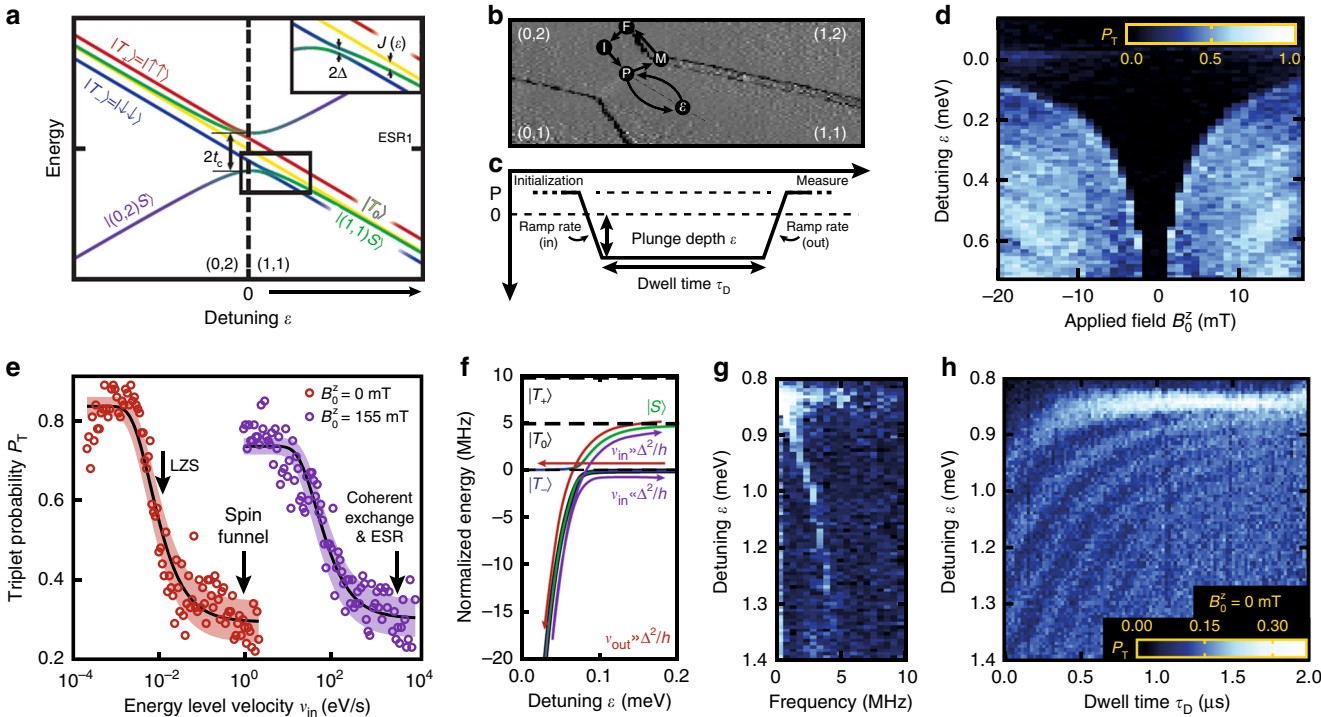

**Fig. 2** Mapping and control of singlet $T_-$ triplet anti-crossing. **a** Energy diagram for the five lowest energy states near the $(0, 2)$–$(1, 1)$ anti-crossing represented in the singlet-triplet basis. **b, c** Five-level pulse sequence used in panels **d**, **e** and **g**. **b** A $|(0, 2)S\rangle$ state is initialised by moving from M, through point F where rapid tunnelling occurs with the reservoir, to point I. From point P, we plunge into the (1,1) region to probe the anti-crossing, and return via P to then move to the latched spin blockade measurement point at M. **c** Plunge depth into (1, 1) between P and $\varepsilon$ as a function of time, illustrating experimental variables including $\varepsilon$ detuning, ramp rates and dwell time. **d** A characteristic spin funnel is observed where the $S_H/T_-$ state degeneracy results in a relaxation hotspot. **e** The $S_H/T_-$ coupling strength $\Delta(\theta)$ is characterized by performing a single passage Landau–Zener excitation experiment[43] at two different $B_0^z$ applied magnetic field settings. Here, the x-axis indicates the rate of change of the $S_H/T_-$ energy separation as extracted from measurements of this energy difference vs. voltage and the voltage ramp rate into (1, 1). The increase of this rate (known as the energy level velocity $\nu$) acts to preserve the $|S_H\rangle$ initial state following the Landau–Zener formula, to which we fit to extract $|\Delta(\theta)|$ (see text). Solid lines show the fit while shaded regions are a 95% confidence interval. Arrows indicate the energy level velocity used for given experiments. **f** Energy diagram representation for the effect of varying ramp rate $\nu_{in}$ with respect to $\Delta$ as in **e** while keeping $\nu_{out}$ diabatic. **g** Fourier transform of **h** Landau–Zener–Stueckelberg interference pattern produced by semi-diabatic double-passage through the $S/T_-$ anti-crossing under zero-field $B^z$ offset

function of small detuning $\varepsilon$ by performing a spin funnel experiment[14]. Here, we initialize in a (0, 2) singlet ground state, $|(0,2)S\rangle$, and pulse toward the spatially separated $|(1, 1)S\rangle$, as shown in Fig. 2b, c. By varying the applied magnetic field $B_0^z$ while dwelling at various values of detuning $\varepsilon$, the location of the anti-crossing can be mapped out via the increased triplet probability $P_T$ (Fig. 2d) due to mixing under $\Delta(\theta)$. Ramping across the anti-crossing causes a coherent population transfer between $|S_H\rangle$ and $|T_-\rangle$ due to Landau–Zener tunnelling[43] proportional to $\exp\left(-2\pi|\Delta(\theta)|^2/\hbar\nu\right)$, characterized by the ratio of $\Delta(\theta)$ to the energy level velocity $\nu = \left|d\left(E_{S_H} - E_{T_-}\right)\right|/dt$. As the ramp rate rises the singlet state $|S_H\rangle$ is increasingly maintained (see Fig. 2f) and so the triplet return probability $P_T$ falls, as we observe in Fig. 2e. By fitting this data (Supplementary Note 4) we estimate $|\Delta(\theta)|$ at the location of the minimum energy gap is $(196 \pm 6)$ kHz at $B_0^z = 0$ (an offset field of $B_{OS}^z = -1.04 \pm 0.06$ mT is estimated from spin funnel asymmetry). Further $|\Delta(\theta)| = 16.72 \pm 1.64$ MHz at $B_0^z = 155$ mT, where the uncertainty here (and elsewhere) corresponds to 95% confidence intervals.

There are a number of possible processes that can contribute to $\Delta$ in the silicon-MOS platform. For 800 ppm nuclear-spin-1/2 $^{29}$Si in the isotopically enriched $^{28}$Si epilayer[44], we expect random hyperfine fields in all vector directions with root-mean-square of order 50 kHz[20] for unpolarized nuclei, so this may contribute to $\Delta$. However, other effects may have a comparable contribution.

At low $B_0^z (\lesssim 50$ mT), Meissner effects from the superconducting aluminum gates can provide transverse local magnetic fields at the location of the QDs; contributions to $\delta B^z$ from this effect of up to a few MHz have been reported[40]. Further, off-diagonal terms in the difference between the electron g-tensors can contribute to coupling between (1, 1) states. Finally, in the presence of inter-dot tunnelling, the interface spin–orbit interaction provides a separate contribution to $\Delta$, leading to estimated couplings of tens of kHz at low-$B_0^z$. Detailed studies on magnetic field magnitude and angle dependence, such as those performed to isolate hyperfine from spin–orbit contributions in nuclear-rich materials such as GaAs[45,46], are required to separate and explore each of these individual effects.

We can further characterize the Hamiltonian in Eq. (1) at much larger detuning $\varepsilon$ than is accessible via the spin funnel by performing a Landau–Zener–Stueckelberg (LZS) interference experiment[18,43] (Fig. 2g, h). This is performed at $B_0^z = 0$, but the residual magnetic field present, which may include some nuclear polarization[47] is sufficient to split the $|T_0\rangle$ and $|T_\pm\rangle$ states. By setting the ramp rate across the $|S_H\rangle/|T_-\rangle$ anti-crossing to $2\pi\nu \approx |\Delta|^2/\hbar$, an approximately equal superposition of both states is created. Dwelling for varying times $\tau_D$ and detunings $\varepsilon$ results in a Stueckelberg phase accumulation $\phi = \int\left(E_{S_H}(\varepsilon[t]) - E_{T_-}(\varepsilon[t])\right)dt/\hbar$, with $E_{S_H}\left(E_{T_-}\right)$ the energy of the $|S_H\rangle(|T_-\rangle)$ state. Depending on the accumulated phase, the

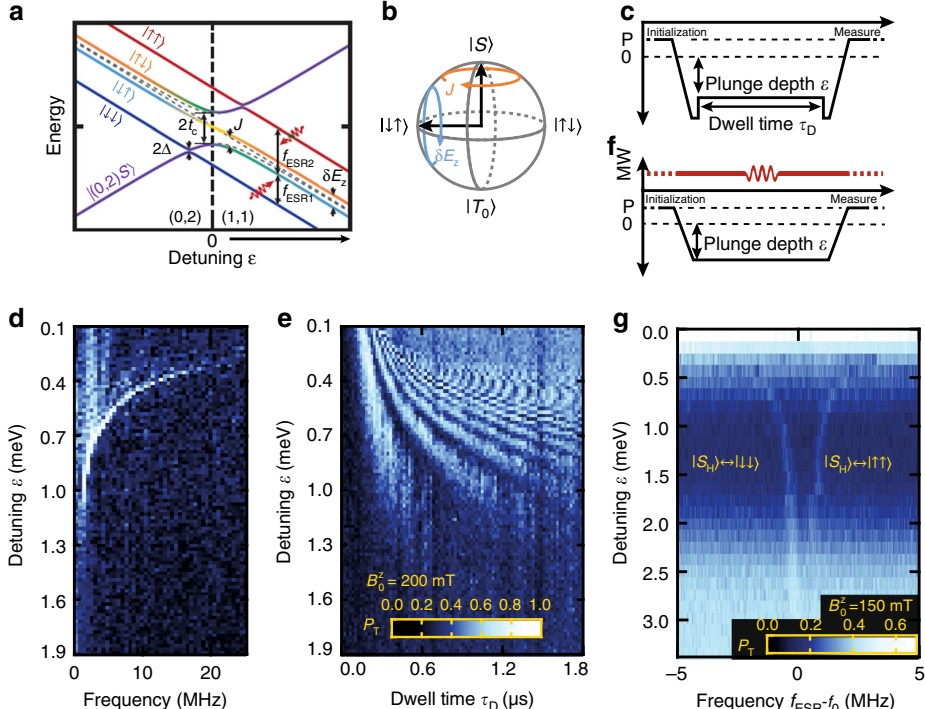

**Fig. 3** Exchange drive oscillations and individual electron ESR at low field. **a** Energy diagram for the five lowest energy states near the (0, 2)–(1, 1) anti-crossing represented in the spin basis. Compared to Fig. 2a, an increased magnetic field $B_0^z$ splits off polarized triplets $|T_\pm\rangle$ while $\delta g$ due to the SO coupling breaks the $|(1,1)S\rangle/|T_0\rangle$ degeneracy producing $\delta E_Z$. **b** Bloch sphere representation of the $|(1,1)S\rangle/|T_0\rangle$ qubit showing effect of the Heisenberg exchange $J$ and $\delta E_Z$ **c**, **d** Coherent Rabi oscillations between $|\downarrow\uparrow\rangle$ and $|\uparrow\downarrow\rangle$ states, driven by exchange $J$. **c** Pulse sequence for data in **d** and **e**; adiabatic ramp (diabatic through the $S_H/T_-$ crossing) prepares $|\downarrow\uparrow\rangle$ (assuming $g_2 > g_1$). Diabatic pulses back to the high exchange region then causes coherent evolution of the state for a period of variable time/depth. The resulting change in population of $|\downarrow\uparrow\rangle$ is mapped back to $|(0,2)S\rangle$ using the inverse adiabatic ramp. **d** Fourier transform of time series **e** which shows exchange driven oscillations between the $|\downarrow\uparrow\rangle$ and $|\uparrow\downarrow\rangle$ states. **f** Pulse sequence used for data in **g**. **g** Triplet probability as a function of detuning $\varepsilon$ and applied ESR frequency with $f_0 = 4.205$ GHz. ESR spin rotations of the spin in the left dot (upper branch) and right dot (lower branch), using an on-chip microwave ESR line. $|\downarrow\uparrow\rangle$ is prepared similar to **b**, a 25 μs ESR pulse of varying frequency is applied rotating $|\downarrow\uparrow\rangle \rightarrow |\uparrow\uparrow\rangle$ when $g_2\mu_B B_0^z/h = f_{ESR}$, and $|\downarrow\uparrow\rangle \rightarrow |\downarrow\downarrow\rangle$, when $g_1\mu_B B_0^z/h = f_{ESR}$; $|\downarrow\uparrow\rangle$ is again mapped back $|(0,2)S\rangle$. We find $|g_2 - g_1| = (0.43 \pm 0.02) \times 10^{-3}$

returning passage through the anti-crossing either constructively interferes, resulting in the blockaded $|T_-\rangle$, or destructively interferes, bringing the system back to $|S_H\rangle$. By keeping $v$ constant throughout the experiment the Fourier transform (Fig. 2g) of the interference pattern (Fig. 2h) directly extracts the energy separation $|E_{S_H}(\varepsilon) - E_{T_-}(\varepsilon)|$ as a function of detuning. In future experiments, this Fourier transform could also be spectroscopically resolved via microwave excitation across the $S_H/T_-$ transitions, either using our integrated ESR control or via photon-assisted tunnelling[48].

We now investigate exchange between the hybridised singlet $|S_H\rangle$ and unpolarized triplet $|T_0\rangle$ by applying an external magnetic field $B_0^z = 200$ mT to strongly split away the $|T_\pm\rangle$ triplet states. At these fields the Zeeman energy difference $\delta E_Z$ dominates exchange $J(\varepsilon)$ deep in the (1, 1) region, and the eigenstates there become $|\downarrow\uparrow\rangle$ and $|\uparrow\downarrow\rangle$, as depicted in Fig. 3a. Maintaining a ramp rate $v$ fast enough to be diabatic with respect to $\Delta$, but slow enough to be adiabatic with respect to $t_c(\varepsilon)$, ensures adiabatic preparation of a ground state $|\downarrow\uparrow\rangle$ or $|\uparrow\downarrow\rangle$, depending upon the sign of $\delta E_Z = g_2\mu_B B_2^z - g_1\mu_B B_1^z$. At $B_0^z = 200$ mT we expect the Meissner effect to be quenched, so that $\delta E_Z$ is dominated by the effective g-factor difference between the dots.

For simplicity we henceforth assume $\delta E_Z > 0$, so that we adiabatically prepare $|\downarrow\uparrow\rangle$ for large $\varepsilon$. Following the pulse sequence illustrated in Fig. 3c, coherent-exchange-driven oscillations can then be observed between $|\downarrow\uparrow\rangle$ and $|\uparrow\downarrow\rangle$ by rapidly plunging the prepared state $|\downarrow\uparrow\rangle$ back towards the (1, 1)–(0, 2)

anti-crossing where $J(\varepsilon)$ is no longer negligible. Variable dwell time $\tau_D$ results in coherent exchange oscillations, and the reversal of the rapid plunge leaves the state in a superposition of $|\downarrow\uparrow\rangle$ and $|\uparrow\downarrow\rangle$. The semi-adiabatic ramp back to (0, 2) maps the final state $|\downarrow\uparrow\rangle$ to the $|(0,2)S\rangle$ singlet, while $|\uparrow\downarrow\rangle$ is mapped to a blockaded state via the $|T_0\rangle$ triplet[14,19]. The resulting data is shown in Fig. 3d, e.

**Individual qubit addressability via electron spin resonance**. We note that previous experiments performed at $B_0^z = 1.4$ T on another SiMOS device exploited the g-factor difference between two QDs in the low-$J(\varepsilon)$ region to perform a two-qubit controlled-phase operation[8]. Utilizing the high-$J(\varepsilon)$ region as above, the $|\downarrow\uparrow\rangle \leftrightarrow |\uparrow\downarrow\rangle$ operation can extend the two-qubit toolbox to include a SWAP gate, with a potentially shorter operation time, in this case with $\tau_{SWAP} \sim 0.25$ μs, limited by exchange pulse rise times.

Having characterized the system in the singlet–triplet basis, we now investigate the compatibility of spin blockade readout with individual QD (i.e., single spin) addressability via electron spin resonance (ESR)[2], a combination desirable for scalable spin qubit architectures incorporating error correction[15,16]. Using the pulse sequence illustrated in Fig. 3f, we again adiabatically prepare the large-$\varepsilon$ ground state $|\downarrow\uparrow\rangle$, as discussed above. We now apply an ac magnetic field to perform ESR with pulse duration 25 μs, supplied by the on-chip microwave transmission line[49] (Fig. 1a), to drive transitions that correspond to $|\downarrow\uparrow\rangle \leftrightarrow |\downarrow\downarrow\rangle$ and

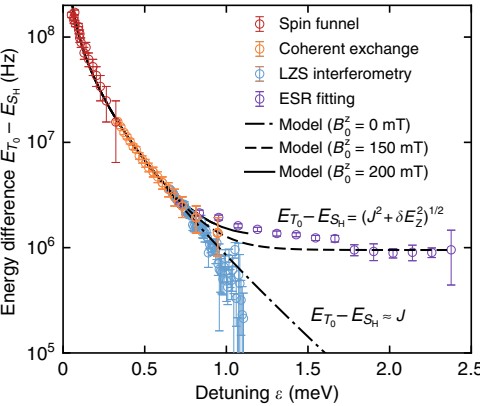

**Fig. 4** Effective exchange with detuning. Exchange energy splitting $\left|E_{S_H}(\varepsilon) - E_{T_0}(\varepsilon)\right|$ as a function of detuning $\varepsilon$, as extracted from the spin-funnel (Similar to Fig. 2d, see Supplementary Note 5 and Supplementary Fig. 4), Landau–Zener–Stueckelberg interferometry (Fig. 2g), coherent exchange oscillations (Fig. 3d) and ESR funnel data (Fig. 3f). Each include 95% confidence intervals based on data fits uncertainties or measurement resolution. The solid/dashed lines represent fits to the data based on the model Hamiltonian in Eq. (1), for which we find $t_c(\varepsilon = 0) = 1.86 \pm 0.03$ GHz

$|\downarrow\uparrow\rangle \leftrightarrow |\uparrow\uparrow\rangle$ at large detuning, when exchange is small (see Fig. 3a). Any excitation from the ground state will now map to the blockaded triplet state population. Figure 3g shows the measured ESR spectrum as a function of detuning $\varepsilon$. The higher frequency $f_{ESR2}$ branch corresponds to a coherent rotation of the electron spin in QD2, while the lower frequency $f_{ESR1}$ rotates the QD1 spin. At large detuning $f_{ESR1} \sim 4.2$ GHz, consistent with the applied magnetic field $B_0^z = 150$ mT for this experiment. As $\varepsilon$ decreases (and $J(\varepsilon)$ increases), the ground state is better described as $|S_H\rangle$, so the transitions become $|S_H\rangle \leftrightarrow |\downarrow\downarrow\rangle$ and $|S_H\rangle \leftrightarrow |\uparrow\uparrow\rangle$ and exchange now competes with ESR, resulting in a lower visibility. For large detuning, the reduction in visibility can be produced from a number of sources. Here, the tuning of the semi-adiabatic ramp rate is critical in this region to prevent excitation (discussed further in Supplementary Note 7). The reduction in visibility seen here is due to an increase in preparation and readout error. This could be mitigated in future experiments though pulse optimisation of the semi-adiabatic ramps.

**Exchange coupling between qubits.** Each of the experiments described above probes the Hamiltonian in Eq. (1) for different ranges of detuning. Figure 4 collates the results of all experiments and plots the energy splitting between the hybridised singlet $|S_H\rangle$ and unpolarised triplet $|T_0\rangle$ across all detuning values. Close to the (0, 2)-(1, 1) anti-crossing, for low $\varepsilon$, the splitting is dominated by exchange coupling $J$, while for large $\varepsilon$, $\delta E_Z$ dominates. As expected, the energy differences obtained from the LZS interferometry (for $B_0^z \approx 0$) diverge from those obtained via ESR (where $B_0^z = 150$ mT), since when $B_0^z \approx 0$ there remains only a small residual $\delta E_Z$ due to combined Meissner screening and weak Overhauser fields. Figure 4 also shows a fit to the data employing the Hamiltonian of Eq. (1) as documented elsewhere. A constant $t_c$ fits poorly; instead a model for a dependence of the tunnel coupling on $\varepsilon$ is employed (see Supplementary Note 6). At the anti-crossing ($\varepsilon = 0$), the curve fit to this model indicates $t_c(\varepsilon = 0) = 1.864 \pm 0.033$ GHz and $\delta g = (0.43 \pm 0.02) \times 10^{-3}$. We note that this tunnel coupling is comparable to that observed for a separate two-qubit device[8] for which $t_c(\varepsilon) = 900\sqrt{2}$ MHz.

## Discussion
By analyzing the error processes present in these experiments, we can identify where improvements will be required before these mechanisms can be integrated into a parity readout tool useful for future multi-qubit architectures. We can discriminate between the effect of various error processes by comparing blockade probability observations under different operating regimes in the exchange oscillation data of Fig. 3e. The histograms shown in Fig. 1g each reveal state preparation and measurement (SPAM)-related errors, leading to a visibility maximum of 98% (orange data) and an error of 0.8% associated with $|(0, 2)S\rangle$ preparation and the transfer process to a latched readout position (red data). Additional to these SPAM errors are the transfer and mapping error processes present when converting states semi-adiabatically from the (0, 2)→(1, 1) or (1, 1)→(0, 2) charge transitions respectively. The combined error from SPAM, state transfer and mapping can be observed from the background visibility at a detuning where exchange is minimal. Here, the prepared $|(0, 2)S\rangle$ state is ideally transferred to and from the (1, 1) region without loss, resulting in zero triplet probability. In contrast, the average blockaded return probability from Fig. 3e (and therefore the combined transfer and mapping errors) saturates to around 30%. From the decay in the oscillations of Fig. 3e as a function of operation time $\tau_D$, we find a maximum control fidelity of $F_\pi = 0.95 \pm 0.04$ at $\varepsilon = 0.6$ meV. We find that the decay time is proportional to the Rabi period, suggesting that exchange noise limits our control fidelity. Further, comparisons with the 61% visibility of the first fringe in Fig. 3e suggests that diabaticity errors due to each fast plunge to/from the exchange position are also present. With respect to our system's utility for providing a parity readout tool, the main error source in the present work appears to occur during adiabatic transfer into and out of the (1, 1) region. Time-dependent simulations (Supplementary Note 7 and Supplementary Table 1) of the model Hamiltonian Eq. (1) show that this error can be well explained by diabaticity with respect to $t_c(\varepsilon)$ near the anti-crossing. We expect that this error can be significantly reduced by optimizing the shape of the ramp as a function of detuning, to remain diabatic with respect to $\Delta$ near the $|S_H\rangle/|T_-\rangle$ crossing, while staying adiabatic elsewhere. Of relevance to the fidelity of exchange-based two-qubit gates, we note that charge and voltage noise will couple via detuning $\varepsilon$ to produce noise in exchange. Our simulations (Supplementary Note 7 and Supplementary Fig. 5) indicate that the level of charge noise expected[7,20] in our system results in a $|S_H\rangle/|T_0\rangle$ oscillation decay consistent with our measurements. The effect of charge noise could be minimized by symmetric biasing[50], with the use of an additional exchange gate.

To conclude, we have for the first time in a silicon device experimentally combined single-spin control using electron spin resonance, with high-fidelity single-shot readout in the singlet–triplet basis. By characterising the relevant energy scales $\Delta$, $\delta E_Z$ and $t_c(\varepsilon)$ of the two-spin Hamiltonian, we found that we could coherently manipulate both the $S/T_-$ and $S/T_0$ states, the latter of which provides potential for a fast two-qubit SWAP gate at high exchange. The integration of low-frequency ESR of individual spins with singlet-triplet based initialisation and readout holds promise for qubit architectures operating at significantly lower magnetic fields and higher temperatures. Future experiments will focus on improvements in operational fidelities, as well as further characterisation of low-frequency ESR operation. The presented initialisation and readout of singlet–triplet states attests to the compatibility of the SiMOS quantum dot platform with parity readout based on spin-blockade, key for the realisation of a future large-scale silicon-based quantum processor[15,16].

                    

## Methods

**Device fabrication**. The device is fabricated on an epitaxially grown, isotopically purified $^{28}$Si epilayer with residual $^{29}$Si concentration of 800 ppm[44]. Following the multi-level gate-stack silicon MOS technology[26], four layers of Al-gates are fabricated on top of a SiO$_2$ dielectric with a thickness of 5.9 nm. Gate layers have a thickness of 25, 60, 80, and 80 nm, with three layers used to form the device and the fourth layer attributed to the ESR transmission line. Overlapping layers are separated by thermally grown aluminum oxide.

**Experimental set-up**. The measurements were conducted in a dilution refrigerator with base temperature $T_{bath} = 30$ mK. DC voltages were applied using battery-powered voltage sources and are combined with voltage pulses using an arbitrary waveform generator (LeCroy ArbStudio 1104) through resistive voltage divider network. Filters were included for DC, slow-pulse and fast-pulse lines (10 Hz to 80 MHz). Microwave pulses were delivered by an Agilent E8257D analogue signal generator, passing signal through a 10 dBm attenuator at the 4 K plate and 3 dBm attenuator at the mixing chamber plate.

All the measured qubit statistics are based on counting the blockade signal in the latched region as described in the main text. The operating region within the experiments involves a system of two tunnel coupled quantum dots with a total of two electrons shared between them. The latched readout procedure involves conditional loading of a third electron from tunnel-coupled reservoir onto one of these dots. Each data point represents the average of between 100 and 1200 single shot blockade events, including experiment trace repetition. Stability maps generated from three level pulsing could be produced with less averaging, with figure data being the average of 40 shots per point.

## Data availability

The data that support the findings of this study are available from the corresponding author upon reasonable request.

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

## Acknowledgements

We thank Mark Gyure for helpful discussions. We acknowledge support from the Australian Research Council (CE11E0001017 and CE170100039), the US Army Research Office (W911NF-13-1-0024 and W911NF-17-1-0198) and the NSW Node of the Australian National Fabrication Facility. The views and conclusions contained in this document are those of the authors and should not be interpreted as representing the official policies, either expressed or implied, of the Army Research Office or the U.S. Government. The U.S. Government is authorized to reproduce and distribute reprints for Government purposes notwithstanding any copyright notation herein. M.V. and B.H. acknowledges support from the Netherlands Organization for Scientific Research (NWO)

through a Rubicon Grant. K.M.I. acknowledges support from a Grant-in-Aid for Scientific Research by MEXT, NanoQuine, FIRST, and the JSPS Core-to-Core Program.

## Author contributions

M.A.F. and B.H. performed experiments. M.V. designed the device, fabricated by K.W.C. and F.E.H. with A.S.D's supervision. K.M.I. prepared and supplied the $^{28}$Si epilayer. W. H., T.T., C.H.Y., and A.L. contributed to the preparation of experiments. M.A.F., B.H., and A.S.D. designed the experiments, with T.D.L., W.H., D.C., K.W.C., T.T., C.H.Y., A.L., A.M. contributing to results discussion and interpretation. M.A.F., B.H., and A.S.D. wrote the manuscript with input from all co-authors.

## Additional information

**Competing interests:** The authors declare no competing interests.

