## [Peer Review File · Nature Communications]

Reviewers' comments:

Reviewer #1 (Remarks to the Author):

The authors demonstrate high-precision control and readout of a double quantum dot spin qubit device in silicon MOS. They show that high-fidelity, single-shot readout based on Pauli spin blockade is fully compatible with single-qubit operations implemented via ESR. This is the first time this has been achieved in a silicon device. They also demonstrate a full range of control and measurement techniques in the same device, including coherent exchange oscillations, Landau-Zener tunneling, latched spin blockade, and Landau-Zener-Stueckelberg interferometry. These techniques are combined to provide a detailed characterization of the two-qubit Hamiltonian.

The fact that all these state-of-the-art techniques can be integrated together in a single device is impressive and constitutes an important milestone for silicon spin qubits, which have risen to the forefront of spin-based quantum computing in recent years. For this reason, I believe the work is appropriate for the broad readership of Nature Communications.

Overall, the manuscript is very well written and the data is clearly presented. I have only two minor modifications/corrections that should be addressed prior to publication:

1. Reference 9 appears to be missing the last author and to have the wrong arxiv number.
2. One point I found a little confusing concerns the labeling of the quantum dots. My understanding is that, e.g., (0,2) describes a charge configuration with 2 electrons in the left dot (i.e., QD2 is the quantum dot that appears on the left in Fig. 1b). I was initially confused by this while reading the section on latched spin blockade. Perhaps the simplest fix would be to add labels QD1 and QD2 in Fig. 1b.

Reviewer #2 (Remarks to the Author):

Review of the manuscript "Integrated silicon qubit platform with single-spin addressability, exchange control and robust single-shot singlet-triplet readout" by M. A. Fogarty et al., submitted for Nature Communications.

In this work, the authors combined various manipulation and characterization methods known for electron spins qubits in double quantum dots (DQD) to a Si-MOS DQD. The used methods Pauli spin blockade, latched Pauli spin blockade, spin funnel, Landau Zener sweeps, Landau-Zener-Stueckelberg oscillations, coherent exchange oscillations, electron spin resonance are not novel (made clear by cited references), but the application of all these methods on one device is new and very interesting for the community. Using these methods the authors determined important parameters of the DQD Hamiltonian in great detail: tunnel coupling(ϵ), singlet-T- coupling, exchange interactions as a function of detuning and the Zeeman energy difference for various magnetic fields across the DQD. In addition, the combination of single spin manipulation and Pauli spin blockade is very timely for quantum error detection (parity read-out) and higher temperature spin qubits. The analysis of all these different methods (SPAM errors, mapping errors, exchange fidelity) is densely written, precise and well completed by the supplements. The results of the different methods are consistent and well discussed at the end. Therefore, I recommend publication of the manuscript in Nature Communications, if the following minor points are addressed:

- The values of ΔE_z and t_c is crucial for the DQD operation. Please comment to what

extend can the g-factor difference be controlled? In what range can $t_c(\epsilon=0)$ be set in the device? Is t_c control independent from g-factor difference control?

- Figure 1C: Please make the zoom-in (surrounded by red rectangle) clearer. At first glance, it appears to be part of the main honeycomb lattice. One then might conclude that the authors observe PSB at the (1,2)-(2,1) transition.

- Add the DQD occupancy numbers to Fig 1d.

- Please add to reference 4 (high-fidelity EDSR): J. Yoneda, Nature Nanotechnology 13, 102 (2018).

- Please add to reference 43 (determination methods for S-T- coupling): F. R. Braakman et al., PRB 89, 075417 (2014).

- Fig 2g, and text: It is worth mentioning that the $S_{H/T}$ - energy splitting (Fourier transform) can be directly measured by applying photon-assisted tunnelling, (one Hamiltonian characterization method not used here) e.g. in L. Schreiber, Nature Communications 2, 556 (2011)

- Fig 3f and text. Comment, why the visibility decreases not only for low epsilon, but also for large detuning.

We thank both Reviewers for the insightful comments made regarding our manuscript. We have reviewed the manuscript and updated it with respect to these comments. We believe that we have addressed the comments put forward by the Reviewers and, as the Reviewers have suggested, the manuscript is now ready to move forward towards publication in *Nature Communications*.

Please find below the detailed response to the reports from each Referee.

Reviewer #1 (Remarks to the Author):

The authors demonstrate high-precision control and readout of a double quantum dot spin qubit device in silicon MOS. They show that high-fidelity, single-shot readout based on Pauli spin blockade is fully compatible with single-qubit operations implemented via ESR. This is the first time this has been achieved in a silicon device. They also demonstrate a full range of control and measurement techniques in the same device, including coherent exchange oscillations, Landau-Zener tunneling, latched spin blockade, and Landau-Zener-Stueckelberg interferometry. These techniques are combined to provide a detailed characterization of the two-qubit Hamiltonian.

The fact that all these state-of-the-art techniques can be integrated together in a single device is impressive and constitutes an important milestone for silicon spin qubits, which have risen to the forefront of spin-based quantum computing in recent years. For this reason, I believe the work is appropriate for the broad readership of Nature Communications.

Response: The Reviewer demonstrates a very clear understanding of the scope and significance of the experiments we present. We thank the Reviewer for their appreciation of our manuscript and their recommendation for publication in *Nature Communications*. We would like to extend our gratitude to the Reviewer for the following comments, the addressing of which will improve the clarity of the experiments to our readers.

Overall, the manuscript is very well written and the data is clearly presented. I have only two minor modifications/corrections that should be addressed prior to publication:

- 1. Reference 9 appears to be missing the last author and to have the wrong arxiv number.*

Response: We thank the Reviewer for bringing this error to our attention. Upon investigation we found that this article has in fact now been published in a peer reviewed journal under the revised title “Resonantly driven CNOT gate for electron spins”. As such, the reference has been updated accordingly.

- 2. One point I found a little confusing concerns the labeling of the quantum dots. My understanding is that, e.g., (0,2) describes a charge configuration with 2 electrons in the left dot (i.e., QD2 is the quantum dot that appears on the left in Fig. 1b). I was initially confused by this while reading the section on latched spin blockade. Perhaps the simplest fix would be to add labels QD1 and QD2 in Fig. 1b.*

Response: The distinction between the dots is a very important detail when describing the readout mechanisms utilized in our manuscript. For this reason, the clear interpretation of which dot is which is important. We thank the Reviewer for this enlightening comment, which indicates that this distinction can be seized more readily by our readers by introducing clear labelling in Fig. 1b. We have introduced this labelling in the specified location, which follows the color convention of the electron Reservoir. We have chosen this as there is a common feature between the Reservoir and the dots, being that they are both based on the two-dimensional electron gas. This coloring (yellow, rather than black or white) has the added benefit of making the labeling of the dots more distinct within this subfigure, which is already quite busy. For convenience, we present Figure 1 below, which compares the original manuscript figure with the revised figure from our resubmission.

Figure 1: Changes to manuscript Fig. 1b. The original subfigure 1b (Left) compared to the revised subfigure (Right). The changes include addition of QD1 and QD2 labeling, and well as an appropriate re-adjustment for the "Reservoir" label.

Reviewer #2 (Remarks to the Author):

In this work, the authors combined various manipulation and characterization methods known for electron spins qubits in double quantum dots (DQD) to a Si-MOS DQD. The used methods Pauli spin blockade, latched Pauli spin blockade, spin funnel, Landau Zener sweeps, Landau-Zener-Stuckelberg oscillations, coherent exchange oscillations, electron spin resonance are not novel (made clear by cited references), but the application of all these methods on one device is new and very interesting for the community. Using these methods the authors determined important parameters of the DQD Hamiltonian in great detail: tunnel coupling(ϵ), singlet-T- coupling, exchange interactions as a function of detuning and the Zeeman energy difference for various magnetic fields across the DQD. In addition, the combination of single spin manipulation and Pauli spin blockade is very timely for quantum error detection (parity read-out) and higher temperature spin qubits. The analysis of all these different methods (SPAM errors, mapping errors, exchange fidelity) is densely written, precise and well completed by the supplements. The results of the different methods are consistent and well discussed at the end. Therefore, I recommend publication of the manuscript in Nature Communications, if the following minor points are addressed:

Response: We thank the Reviewer for their helpful comments on our manuscript and their endorsement for publication in *Nature Communications*, conditional on our satisfactory corrections. We address each of these comments below:

1. *The values of δE_Z and t_c is crucial for the DQD operation. Please comment to what extend can the g -factor difference be controlled? In what range can $t_c(\epsilon = 0)$ be set in the device? Is t_c control independent from g -factor difference control?*

Response: The Reviewer makes the astute observation that performance and control of these experiments are reliant upon the underlying Hamiltonian parameters δE_Z and t_c . The Reviewer suggests that it would therefore be prudent to describe and discuss how these two parameters are defined and/or controlled in the SiMOS quantum dot qubit platform.

As the Reviewer comments, control over the g -factor difference between dots gives tunability of the δE_Z term. In our experiments, this value is intrinsically tied to the external magnetic field settings. A recent study has shown that both the magnitude and direction of this applied field can strongly modulate this term. This is summarised by the following addition to the manuscript text:

Page 3, column 2, paragraph 1 (Middle):

For high in-plane magnetic field, the varying effective g -factors result from a combination of interface spin-orbit terms, which depend on local strain, electric fields, and the atomistic details of the oxide interface\cite{Yang2012,Veldhorst2015a,Ferdous2017}. “Further, recent studies have shown that this Hamiltonian parameter can be further modulated by the direction of this applied field with respect to the crystallographic axis\cite{Jock2017}.”

Page 3, column 2, paragraph 1 (End):

“For these experiments, the scale of δE_Z is predominantly set by choice in the magnitude of the external magnetic field; however, inevitable microscopic dot-to-dot variation in this

parameter for future qubit arrays would best be handled by calibration and refocusing methods\cite{Jones2016}.”

The other Hamiltonian parameter discussed is the tunnel coupling t_c . For the designs presented here and elsewhere, this parameter is fixed; set by the device geometry. In order to actively control this parameter within experiments such as these, significant changes to the architecture needs to be made in order to implement what is known as an “exchange gate” (one which has the sole purpose of electrostatically controlling t_c).

We have summarised this discussion in our revised manuscript by introducing the following text:

Page 3, column 2, paragraph 2 (middle):

In previous experiments\cite{Veldhorst2015} on a similar SiMOS two-qubit device the tunnel coupling at the anti-crossing was estimated as $900\sqrt{2}$ MHz. “For both devices, t_c at the anti-crossing is fixed, set by the device geometry. This parameter can be made tunable via the incorporation of exchange gates\cite{Loss1998} into the SiMOS architecture.”

We would like to thank the Reviewer for the suggested clarifications on these subjects. With regards to the Reviewer’s final comment, we note that changing t_c at the anti-crossing via additional exchange gates would not be expected to drastically change the g -factor variation. However, such a modification would vary the electric fields across the device, possibly leading to differing mean values of g -factor and to different sensitivities to changes in gate voltages. We appreciate the Reviewer’s observation; however we do not anticipate that these minor variations in g -factor would have substantial impact on the way this multidot system ultimately performs, and therefore believe that further discussing this speculative complication in the main text would be unnecessarily distracting for the reader.

2. *Figure 1C: Please make the zoom-in (surrounded by red rectangle) clearer. At first glance, it appears to be part of the main honeycomb lattice. One then might conclude that the authors observe PSB at the (1,2)-(2,1) transition.*

Response: We thank the Reviewer for bringing the potential for misreading this figure to our attention. Indeed, as the corner of the inset box is aligned very closely to the (1,1)-(1,2) transition, one could misinterpret the inset data as additional charge occupancy transitions as the Reviewer has suggested. To rectify this issue, we have modified the figure to accentuate the presence of the inset. We believe that this has been achieved with satisfactory emphasis by offsetting the inset from the boarder of Fig. 1c. In addition, a small degree of shadowing beneath the inset box allows it to appear as if it is popping out from the main data, leaving little room for interpreting the inset as a part of the main stability diagram. These changes can be observed in in Figure 2 of this response letter.

Figure 2: Changes to manuscript Figure 1c&d. The original subfigures 1c&d (Left) compared to the revised subfigures (Right). Changes include the emphasis surrounding the Fig. 1c inset as well as the addition of charge occupancy labels in Fig. 1d.

3. Add the DQD occupancy numbers to Fig 1d.

Response: We thank the Reviewer for pointing out this oversight. We have added these occupancy numbers to Fig. 1d of the revised manuscript which can be observed in Figure 2 of this response letter.

4. Please add to reference 4 (high-fidelity EDSR): J. Yoneda, *Nature Nanotechnology* 13, 102 (2018).

Response: We have added this reference, which is a recent publication demonstrating high fidelity gates produced with EDSR. It is indeed an appropriate reference to add to this discussion as it demonstrates the current state-of-the-art.

5. Please add to reference 43 (determination methods for S-T-coupling): F. R. Braakman et al., *PRB* 89, 075417 (2014).

Response: We have added this reference, which utilises photon assisted tunnelling to extract coupling energy between the singlet and polarized triplets, to our manuscript. This reference is indeed appropriate as it demonstrates a method for Hamiltonian characterization, which has not been applied in our manuscript. We thank the Reviewer for this suggestion.

6. Fig 2g, and text: It is worth mentioning that the S_H / T_- energy splitting (Fourier transform) can be directly measured by applying photon-assisted tunnelling, (one Hamiltonian characterization method not used here) e.g. in L. Schreiber, *Nature Communications* 2, 556 (2011)

Response: This comment is actually also related to the previous suggestion from the Reviewer. Photon assisted tunnelling is indeed a complimentary method through which the presence and magnitude of the coupling between the singlet and polarized triplet states have been assessed. Therefore, we have also included the reference suggested by the Reviewer within our discussion on the magnitude and origin of this coupling energy. The reference suggested by the Reviewer is a welcome inclusion into our manuscript as a means of introducing this family of experiments as an alternative approach to our methods.

To accommodate this discussion, we have made the following changes:

Page 5, Column 1, Paragraph 2 (end):

“In future experiments, this Fourier transform could also be spectroscopically resolved via microwave excitation across the S/T- transitions, either using our integrated ESR control or via photon assisted tunnelling\cite{Schreiber2011}.”

7. *Fig 3f and text. Comment, why the visibility decreases not only for low epsilon, but also for large detuning.*

Response: The region which the Reviewer refers to is an area where there is an increase in the background triplet probability which results in the loss of visibility of the ESR spectrum branches. In order to communicate this to our readers, we have added the following discussion, including a suggestion for how this might be rectified in the future:

Page 5, Column 1, Paragraph 3 (end):

“For large detuning, the ramp rate of semi-adiabatic passage is critical in order to prevent excitation (discussed further in Supplementary Information S6). The reduction in visibility seen here is due to an increase in preparation and readout error. This could be mitigated in future experiments though pulse optimisation of the semi-adiabatic ramps.”

REVIEWERS' COMMENTS:

Reviewer #2 (Remarks to the Author):

Review of the revised manuscript "Integrated silicon qubit platform with single-spin addressability, exchange control and robust single-shot singlet-triplet readout" by M. A. Fogarty et al., submitted for Nature Communications.

The authors well addressed all items of my review report. Therefore, I recommend the publication of the revised manuscript in Nature Communications.